# Marine deforestation leads to widespread loss of ecosystem function

**Matthew Edwards**[1]*, **Brenda Konar**[2], **Ju-Hyoung Kim**[3], **Scott Gabara**[1,4],
**Genoa Sullaway**[1], **Tristin McHugh**[1], **Michael Spector**[1], **Sadie Small**[1]

**1** Department of Biology, San Diego State University, San Diego, CA, United States of America, **2** College of Fisheries and Ocean Sciences, University of Alaska Fairbanks, Fairbanks, Alaska, United States of America, **3** Marine Applied Biosciences, Kunsan National University, Gunsan, South Korea, **4** Department of Environmental Science and Policy, University of California, Davis, California, United States of America

* medwards@sdsu.edu

**Data Availability Statement:** Data are available on our NSF bco-dmo data page at https://www.bco-dmo.org/dataset/755658

**Funding:** This research was funded by grants from the National Science Foundation (OCE1435194) to

## Abstract

Trophic interactions can result in changes to the abundance and distribution of habitat-forming species that dramatically reduce ecosystem functioning. In the coastal zone of the Aleutian Archipelago, overgrazing by herbivorous sea urchins that began in the 1990s resulted in widespread deforestation of the region's kelp forests, which led to lower macroalgal abundances and higher benthic irradiances. We examined how this deforestation impacted ecosystem function by comparing patterns of net ecosystem production (*NEP*), gross primary production (*GPP*), ecosystem respiration (*Re*), and the range between *GPP* and *Re* in remnant kelp forests, urchin barrens, and habitats that were in transition between the two habitat types at nine islands that spanned more than 1000 kilometers of the archipelago. Our results show that deforestation, on average, resulted in a 24% reduction in *GPP*, a 26% reduction in *Re*, and a 24% reduction in the range between *GPP* and *Re*. Further, the transition habitats were intermediate to the kelp forests and urchin barrens for these metrics. These opposing metabolic processes remained in balance; however, which resulted in little-to-no changes to *NEP*. These effects of deforestation on ecosystem productivity, however, were highly variable between years and among the study islands. In light of the worldwide declines in kelp forests observed in recent decades, our findings suggest that marine deforestation profoundly affects how coastal ecosystems function.

## Introduction

Consumers fundamentally affect ecosystems through trophic interactions [1]. These interactions are especially important if they result in changes to the abundance or distribution of ecosystem engineers, such as forest-forming trees, which can lead to changes in microclimates, biodiversity, primary production, nutrient cycling, and energy flow [2]. For example, the reintroduction of gray wolves (*Canis lupus*) into Yellowstone National Park, USA in the 1990s resulted in increased predation on elk (*Cervus elaphus*) and subsequently reduced herbivory on canopy-forming trees such as aspens (*Populus tremuloides*), willows (*Salix* spp.), and cottonwoods (*Populus* spp.) [3]. This ultimately led to changes in the morphology and hydrology

ME and (OCE1435205) to BK, and the National Research Foundation (NRF-2018R1C1B6008523 and NRF-2015R1C1A1A01054831) to JHK.

**Competing interests:** NO - Authors have no competing interests.

of the region's river systems and its riparian plant communities [4,5]. Similarly, large marine algae, such as kelps, can form subtidal forests whose biogenic structures alter hydrodynamic, nutrient and light conditions, modify patterns of biodiversity, enhance primary production and carbon sequestration, and provide food and habitat for numerous other species [6–9]. Consequently, the loss of these forest-forming kelps and the benthic communities they support can have dramatic impacts to how nearshore ecosystems function, especially if they occur over large geographic areas. Indeed, kelp deforestation has occurred in numerous areas worldwide in recent decades due to a variety of forcing factors [10,11], and the subtidal rocky reefs of the Aleutian Archipelago serve as a model system to investigate the broader impacts of such deforestation. These forests have historically been dominated by dense populations of the surface canopy-forming kelp *Eualaria fistulosa*, several species of understory kelps such as *Laminaria* spp. and *Agarum* spp., the brown alga *Desmarestia* spp., and numerous species of fleshy read algae. However, the collapse of sea otter (*Enhydra lutris*) populations led to large increases in their primary prey, herbivorous sea urchins (*Strongylocentrotus polyacanthus*), which subsequently resulted in overgrazing and widespread losses of the region's kelp forests [12]. This collapse began in the late 1990s, likely in response to a dietary shift by killer whales toward sea otters, and by 2000 sea otter densities had declined throughout the archipelago to around 5–10% of their estimated equilibrium density [13]. Currently, sea otters are largely absent from or are in very low abundances on many of the islands and most of the kelp forests have either disappeared from the archipelago or are in the process of disappearing, although some small forests remain in their 'historical state' at scattered locations on most of the islands [14,15] (Fig 1). These remnant forests provide a valuable benchmark against which we evaluated the effects of widespread deforestation on an important metric of ecosystem function, namely primary productivity.

Characterizing patterns of biodiversity and primary productivity is essential to fully understanding ecosystem function [16,17]. The latter includes three basic metrics: gross primary production (*GPP*), which describes all the $CO_2$ fixed by the autotrophs during photosynthesis, total ecosystem respiration (*Re*), which describes the release of $CO_2$ during the production of energy by autotrophs, heterotrophs, decomposers and microbes, and net ecosystem production (*NEP*), which is the difference between *GPP* and *Re* and describes net changes in the total amount of organic carbon in an ecosystem available for consumption, storage and export to adjacent ecosystems, or nonbiological oxidation to carbon dioxide [18–21]. In general, ecosystems with high rates of *GPP* also exhibit high rates of *Re*, with the central tendency being that *GPP* and *Re* are in balance (i.e., similar in magnitude) and therefore have median *GPP* / *Re* ratios close to 1.0, and *NEP* values near zero [21,22]. Indeed, a review of five decades (1950 to 1990) of studies in aquatic ecosystems demonstrated that these two opposing processes are generally in balance, although unproductive ecosystems tend towards net heterotrophy with *GPP* / *Re* < 1.0 and *NEP* < 0, while productive ecosystems tend towards net autotrophy with *GPP* / *Re* > 1.0 and *NEP* > 0 [21,22]. Further, the amount of *Re* associated with any given *GPP* in shallow coastal ecosystems tends to be greater when the complete benthic communities are considered [22]. This may be especially true if microbial metabolism, which is an important component of *Re*, is large compared to *GPP* [20–22]. This is important for coastal kelp forests, which host a higher diversity of microbes relative to the adjacent ocean waters [23–27]. Consequently, loss of these forests may lead to complex patterns of *GPP*, *Re*, and *NEP* within coastal ecosystems. On one hand, reductions in primary producer biomass should result in lowered *GPP* and thus reduced *NEP*. Alternately, deforestation may result in lowered abundances of invertebrates, fishes and microbes, which may lead to reduced *Re* and thus enhanced *NEP*. At the same time, loss of the habitat-forming kelps also results in elevated benthic irradiances (measured as photosynthetically active radiation (*PAR*) [18] and thus potentially to enhanced

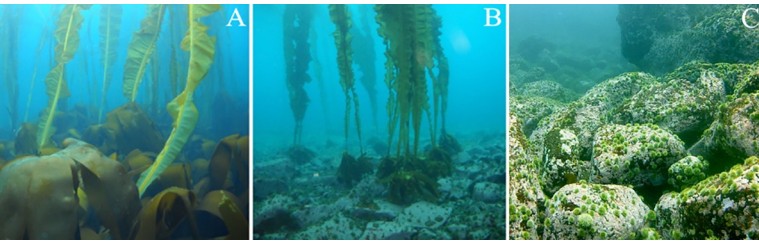

**Fig 1. Three habitat types.** Photographs of each habitat type showing (A) high abundance of benthic macroalgae and canopy-forming kelps in the kelp forests, (B) lack of benthic macroalgae but remaining canopy-forming kelps and high abundances of sea urchins in the transition habitats, and (C) lack of benthic macroalgae and canopy-forming kelps, but high abundances of sea urchins in the urchin barrens.

compensatory production by any remaining fleshy macroalgae, encrusting coralline algae, and microalgae [28–30], which can result in greater *NEP*. Thus, understanding how *GPP*, *Re*, and *NEP* change with kelp forest change can be instrumental in discerning the broader impacts of deforestation on ecosystem productivity. This may be especially relevant for the Aleutian Archipelago where widespread kelp deforestation has resulted in significant reductions in fishes, invertebrates and fleshy macroalgae, increases in the exposure of encrusting coralline algae [12,31], and elevated benthic irradiances [14].

## Results

We used benthic chambers to study patterns of *GPP*, *Re*, and *NEP* within remnant kelp forests, urchin barrens, and habitats that were in transition to becoming urchin barrens (i.e., they had lost all benthic fleshy macroalgae but still had abundant stands of the canopy-forming *Eualaria fistulosa*; Fig 1) at nine islands spanning more than 1000 kilometers of the Aleutian Archipelago (Fig 2, Table 1). Kelp forests and urchin barrens occur as alternate stable states of one another, often with sharply delineated boundaries between them, and exhibit little-to-no overlap in community assemblages [15,33] (Fig 1). Indeed, the benthic communities within our chambers reflected these assemblages, with the chambers deployed in the kelp forests having more than a 10-fold greater biomass of fleshy macroalgae, which were predominantly stipitate kelps, than those deployed in the urchin barrens, and the chambers deployed in the urchin barrens having a nearly 3-fold greater biomass of urchins than those deployed in the kelp forests (Figs 3 and 4). The chambers deployed within transition habitats contained high abundances of urchins and little-to-no fleshy macroalgae, except for the canopy-forming *E. fistulosa*. The chambers within all three habitats had high bottom covers of encrusting coralline algae below the fleshy macroalgae, which became exposed following deforestation. Benthic irradiances, measured as photosynthetically active radiation (*PAR*), varied among the three habitat types (ANOVA: $F_{2,14} = 4.826$, p = 0.025), but this was variable among the nine islands and two study years (Habitat*Island(Year) interaction: $F_{14,33} = 4.426$, p <0.001; Table 2). Generally, *PAR* was greatest in the urchin barrens, lowest in the kelp forests, and intermediate in the transition habitats (Fig 5).

We examined how *GPP*, *Re* and *NEP*, and the balance between *GPP* and *Re* differed among the habitat types by measuring changes in seawater oxygen concentrations within replicate (n = 3) chambers (collapsible benthic incubation tents; hereafter cBITs) that were placed on the benthos over representative assemblages within each habitat type at each island. We predicted that *NEP* at the benthos would be reduced in the urchin barrens due to the loss of photosynthetic macroalgae. Instead, we found that *NEP* did not differ between the habitat types ANOVA: $F_{2,14} = 0.530$, p = 0.600), nor did it differ from zero (i.e., *GPP* = *Re*) in any of the

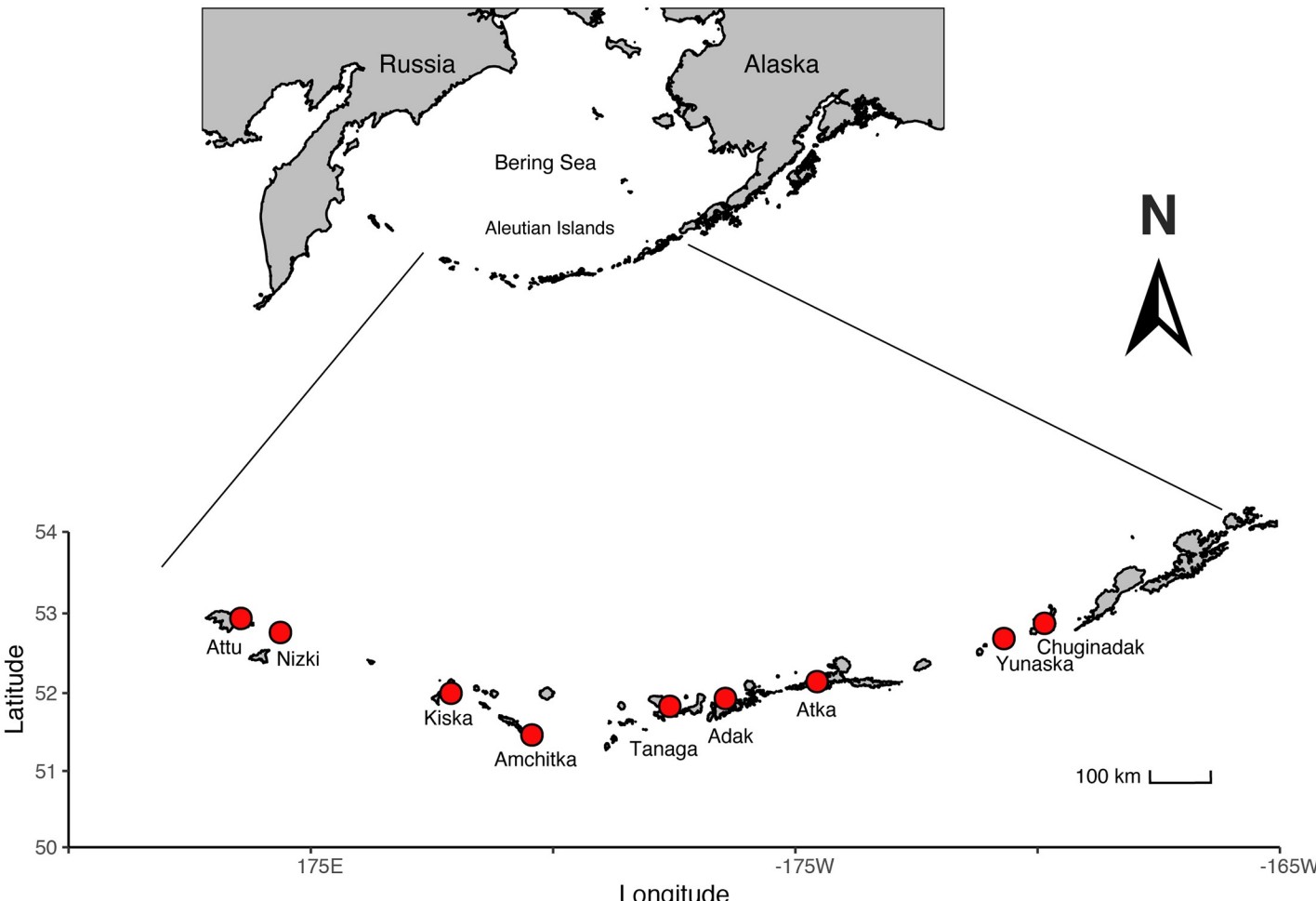

**Fig 2. Map of the Aleutian Archipelago.** Map of the Aleutian Archipelago showing locations of the nine islands (denoted by red circles) where ecosystem productivity (*NEP*, *GPP* and *Re*) was measured in the cBITs. Shoreline data was obtained from the Global Self-Consistent Hierarchical High-resolution Shoreline (GSHHG) dataset version 2.3.4 (www.soest.hawaii.edu/wessel/gshhg/) [32].

habitat types (Table 2, Figs 6 and 7). The effects of habitat type on *NEP*, however, varied among the nine islands visited during the two study years (Habitat*Island(Year) interaction: $F_{14,41}$, = 4.541, p < 0.001; Table 2). Specifically, *NEP* was greater in the urchin barrens than in the kelp forests on five of the islands and lower in the urchin barrens on three of the islands, with the average difference being 25.58 ± 373.26 mg $O^2$ $m^{-2}$ $day^{-1}$ lower in the urchin barrens (i.e., the deforested habitats) (Tables 3 and 4). The change on one island (Attu) was not determined due to lost replication (Tables 1 and 4). However, when averaged across all nine islands, *NEP* was generally lowest (-239.73 ± 425.16 mg $O^2$ $m^{-2}$ $day^{-1}$, mean ± SE) in the kelp forests, highest (-59.60 ± 145.32 mg $O^2$ $m^{-2}$ $day^{-1}$) in the transition habitats, and intermediate (-120.08 ± 338.07 mg $O^2$ $m^{-2}$ $day^{-1}$) in the urchin barrens (Table 3, Fig 4). Benthic *GPP* also did not vary among the habitat types (ANOVA: $F_{2, 14}$ = 0.234, p = 0.794), but when averaged across islands, *GPP* was highest in the kelp forests (1,806.14 ± 521.75 mg $O^2$ $m^{-2}$ $day^{-1}$; mean ± SE), lowest in the urchin barrens (1,367.77 ± 483.99 mg $O^2$ $m^{-2}$ $day^{-1}$), and intermediate in the transition habitats (1,494.22 ± 452.41 mg $O^2$ $m^{-2}$ $day^{-1}$) (Fig 4; Table 3). Like *NEP*, the effects of habitat type varied among the nine islands visited in the two study years (Habitat*Island(Year) interactions: $F_{14,41}$ = 2.166, p = 0.028; Table 2). Specifically, *GPP* was lower in

**Table 1. List of the nine islands in the Aleutian Archipelago where cBITs were deployed to measure *NEP*, *GPP* and *Re* during 2016 and 2017, and the six islands where all macroalgae and invertebrates were collected from within the cBITs to estimate their biomass during 2016.** The number cBITs deployed, the deployment year, and whether macroalgae and invertebrates were collected from within the cBITs at each island are noted.

| Island | Year | No. cBITs deployed | | | Collections made? |
|---|---|---|---|---|---|
| | | Kelp forests | Transition habitats | Urchin barrens | |
| Adak | 2016 | 3 | 2 | 3 | No |
| Amchitka | 2017 | 3 | 2 | 3 | Yes |
| Atka | 2017 | 3 | 3 | 2 | Yes |
| Attu | 2017 | 3 | 3 | 1 | Yes |
| Chuginadak | 2016 | 2 | 2 | 3 | Yes |
| Kiska | 2017 | 3 | 3 | 2 | Yes |
| Nizki | 2017 | 3 | 3 | 2 | Yes |
| Tanaga | 2016 | 2 | 1 | 2 | No |
| Yunaska | 2017 | 3 | 3 | 3 | No |
| **Totals** | | **25** | **22** | **21** | **6** |

the urchin barrens than in the kelp forests on all but two of the islands, by an average of $461.60 \pm 578.69$ mg $O_2$ m$^{-2}$ day$^{-1}$ (mean ± SE) (Table 4). *Re* also did not vary among the habitat types ($F_{2,14} = 0.390$, $p = 0.684$), but when averaged across all nine islands, *Re* was again highest in the kelp forests ($1,994.91 \pm 574.11$ mg $O_2$ m$^{-2}$ day$^{-1}$), lowest in the urchin barrens ($1,474.51 \pm 546.83$ mg $O_2$ m$^{-2}$ day$^{-1}$), and intermediate in the transition habitats ($1,553.84 \pm 469.81$ mg $O_2$ m$^{-2}$

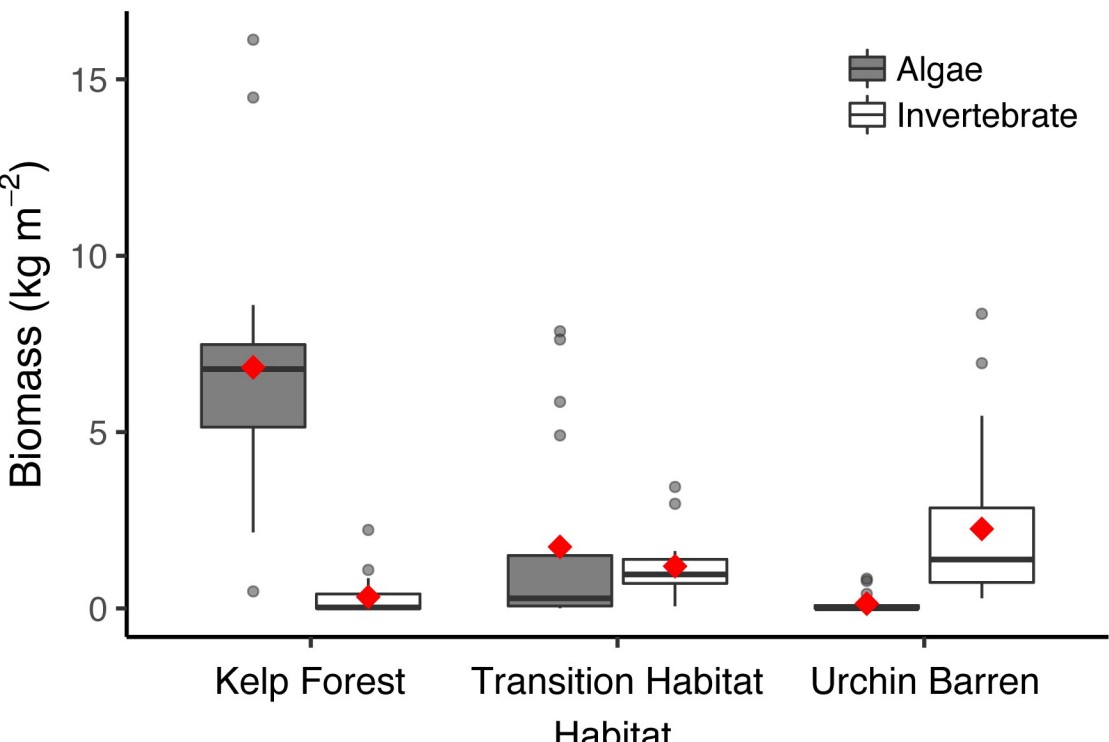

**Fig 3. Algae and invertebrate biomass.** Box plots showing (A) Macroalgae (gray bars) and invertebrate (white bars) biomass measured in the cBITs deployed within each habitat type (kelp forests, transition habitats, and urchin barrens) at six islands during 2017 (Table 1). Red diamonds represent mean values, and horizontal lines represent median values.

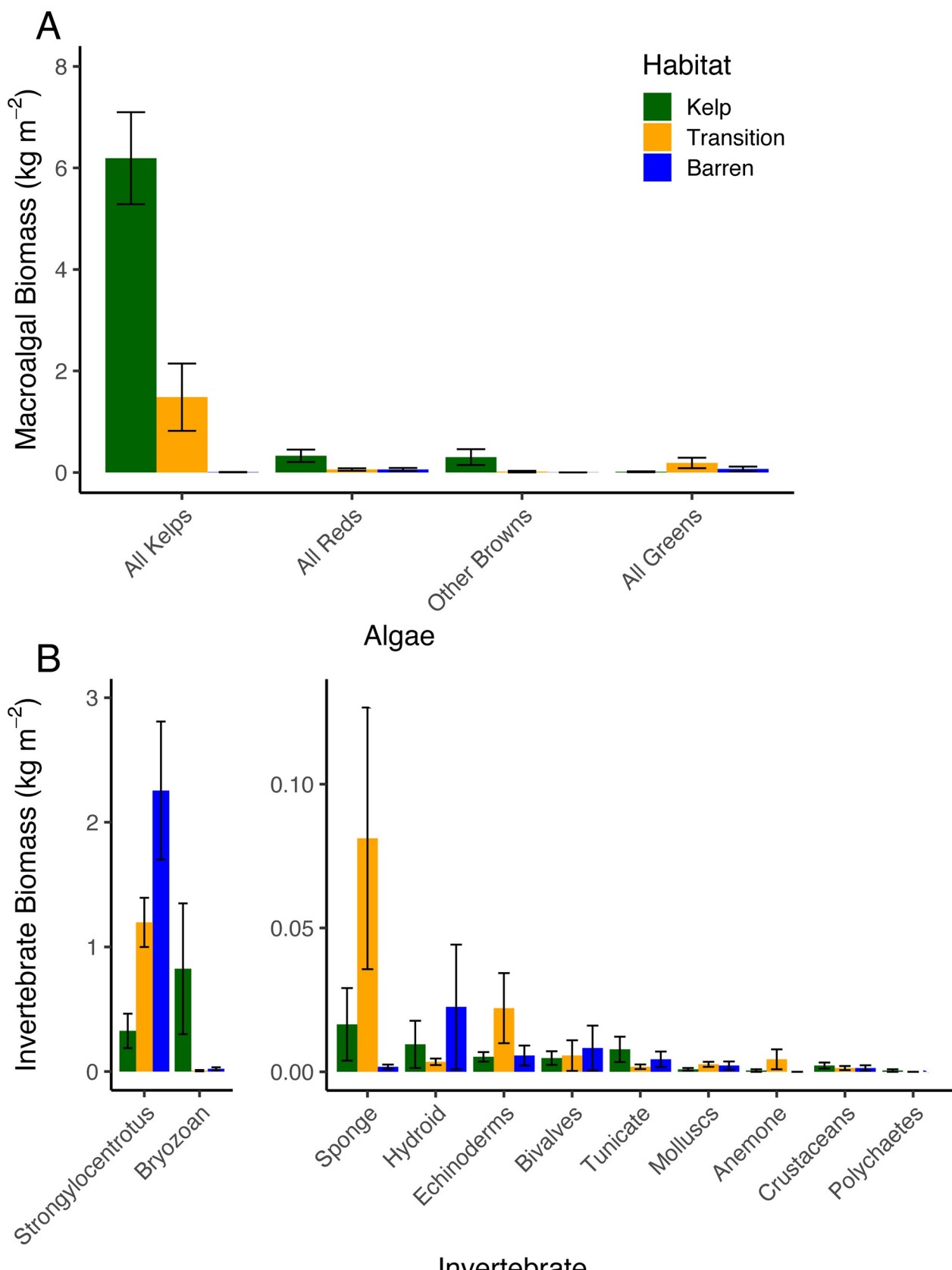

**Fig 4. Algae and invertebrate biomass.** Mean biomass (± SE) of (A) all kelps, and red, brown and green macroalgae, and (B) the most abundant taxonomic groups of invertebrates collected from within the cBITs in each habitat type at six of the islands where the cBITs were deployed in 2017 (Table 1). Fig 5B is divided into two panels, with abundant taxa on the left panel, and rarer taxa on the right panel.

**Table 2. Results of separate three-way nested analyses of variance testing for differences in A) net ecosystem production (*NEP*), B) gross primary production (*GPP*), C) ecosystem respiration (*Re*), D) the range between *GPP* and *Re*, and E) irradiance (*PAR*) among the two sample years, nine islands, and three habitat types (kelp forests, transition habitats, and urchin barrens).** For each analysis, year and habitat type were fixed factors, and island nested within year was a random factor. The model $r^2$ is given for each analysis.

**A) *NEP* ($r^2 = 0.74$)**

| Source | Type III SS | df | Mean Squares | F-ratio | p-value |
|---|---|---|---|---|---|
| Year | 146.911 | 1 | 146.911 | 8.319 | 0.006 |
| Habitat | 85.076 | 2 | 42.538 | 0.53 | 0.6 |
| Habitat*Year | 365.04 | 2 | 182.52 | 10.336 | <0.001 |
| Island(Year) | 610.045 | 7 | 87.149 | 4.935 | <0.001 |
| Habitat*Island(Year) | 1122.729 | 14 | 80.195 | 4.541 | <0.001 |
| Error | 724.009 | 41 | 17.659 | | |

**B) *GPP* ($r^2 = 0.72$)**

| Source | Type III SS | df | Mean Squares | F-ratio | p-value |
|---|---|---|---|---|---|
| Year | 1060.514 | 1 | 1060.514 | 15 | <0.001 |
| Habitat | 71.658 | 2 | 35.829 | 0.234 | 0.794 |
| Habitat*Year | 416.847 | 2 | 208.424 | 2.948 | 0.064 |
| Island(Year) | 3497.967 | 7 | 499.71 | 7.068 | <0.001 |
| Habitat*Island(Year) | 2144.29 | 14 | 153.164 | 2.166 | 0.028 |
| Error | 2898.811 | 41 | 70.703 | | |

**C) *Re* ($r^2 = 0.78$)**

| Source | Type III SS | df | Mean Squares | F-ratio | p-value |
|---|---|---|---|---|---|
| Year | 1456.821 | 1 | 1456.821 | 24.825 | <0.001 |
| Habitat | 125.51 | 2 | 62.755 | 0.39 | 0.684 |
| Habitat*Year | 946.319 | 2 | 473.16 | 8.063 | 0.001 |
| Island(Year) | 3081.525 | 7 | 440.218 | 7.501 | <0.001 |
| Habitat*Island(Year) | 2254.327 | 14 | 161.023 | 2.744 | 0.006 |
| Error | 2406.048 | 41 | 58.684 | | |

**D) *Range* ($r^2 = 0.75$)**

| Source | Type III SS | df | Mean Squares | F-ratio | p-value |
|---|---|---|---|---|---|
| Year | 2503.758 | 1 | 2503.758 | 20.237 | <0.001 |
| Habitat | 184.788 | 2 | 92.394 | 0.318 | 0.733 |
| Habitat*Year | 1267.372 | 2 | 633.686 | 5.122 | 0.01 |
| Island(Year) | 6204.846 | 7 | 886.407 | 7.164 | <0.001 |
| Habitat*Island(Year) | 4064.628 | 14 | 290.331 | 2.347 | 0.017 |
| Error | 5072.698 | 41 | 123.724 | | |

**E) *PAR* ($r^2 = 0.83$)**

| Source | Type III SS | df | Mean Squares | F-ratio | p-value |
|---|---|---|---|---|---|
| Year | 0.898 | 1 | 0.898 | 8.503 | 0.006 |
| Habitat | 4.507 | 2 | 2.254 | 4.826 | 0.025 |
| Habitat*Year | 1.231 | 2 | 0.616 | 5.832 | 0.007 |
| Island(Year) | 2.066 | 7 | 0.295 | 2.795 | 0.021 |
| Habitat*Island(Year) | 6.542 | 14 | 0.467 | 4.426 | <0.001 |

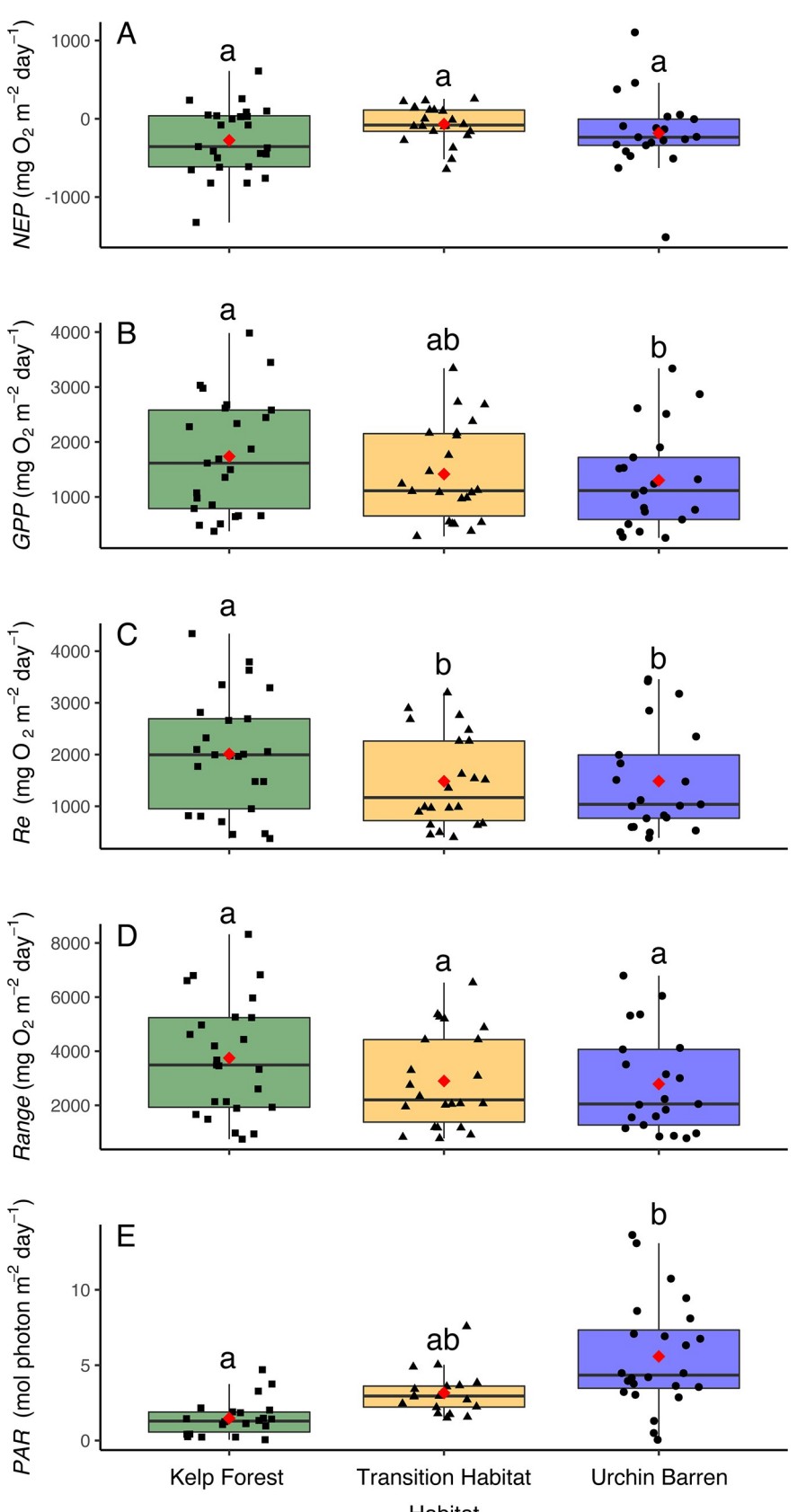

**Fig 5. Production metrics.** Box plots showing (A) Net Ecosystem Production (*NEP*), (B) Gross Primary Production (*GPP*), (C) Ecosystem Respiration (*Re*), (D) the range between *GPP* and *Re* (*Range*), and (E) Irradiance (*PAR*), as measured in the cBITs deployed within each habitat type (kelp forests, transition habitats, and urchin barrens) at nine islands during 2016 and 2017 (Fig 2, Table 1). Red diamonds represent mean values, and horizontal lines represent median values. Boxes within each graph that do not share letters represent significant differences between habitat pairs.

day$^{-1}$) (Fig 4; Table 3). As with *NEP* and *GPP*, the effects of habitat type varied among the nine islands visited in the two study years (Habitat*Island(Year) interactions: $F_{2,14}$ = 2.744, p = 0.006; Table 2). Specifically, *Re* was lower in the urchin barrens than in the kelp forests on four of the islands and greater in the urchin barrens on four of the islands, with the average difference being 472.09 ± 734.70 mg O$^2$ m$^{-2}$ day$^{-1}$ lower in the urchin barrens (Table 4). Lastly, the range between *GPP* and *Re*, which we believe to be a better measure of ecosystem function regarding productivity than *NEP* alone, did not differ among the habitat types (ANOVA: $F_{2,14}$ = 0.318, p = 0.733), but was again greatest in the kelp forests (3,750 ± 1,069.01 mg O$^2$ m$^{-2}$ day$^{-1}$), lowest in the urchin barrens (2,860.94 ± 994.44 mg O$^2$ m$^{-2}$ day$^{-1}$), and intermediate in the transition habitats (3,047.98 ± 910.36 mg O$^2$ m$^{-2}$ day$^{-1}$) (Table 3, Fig 4). This again varied among the study islands visited in the two study years (Habitat*Island(Year) interactions: $F_{14,41}$ = 2.347, p = 0.017; Table 2). Specifically, the range between *GPP* and *Re* was lower in the urchin barrens than in the kelp forests on five of the islands and greater in the urchin barrens on two of the islands, with the average difference being 933.69 ± 1,262.65 mg O$^2$ m$^{-2}$ day$^{-1}$ lower in the urchin barrens (Table 4).

Although the effects of deforestation on all three metrics of productivity varied among the islands visited in the two study years, some general patterns were evident. When considered across all nine islands, *GPP, Re* and the range between *GPP* and *Re* were each greatest in the kelp forests, intermediate in the transition habitats, and lowest in the urchin barrens. Specifically, *GPP* was 24% higher, on average, in the kelp forests than in the urchin barrens, and 17% higher, on average, in the kelp forests than in the transition habitats, but it differed by only 7% between the transition habitats and urchin barrens (Table 3, Fig 4). Benthic *Re* was 26% higher, on average, in the kelp forests than in the urchin barrens, and 22% higher in the kelp forest than the transition habitats, but it differed by less than 1% between the transition habitats and

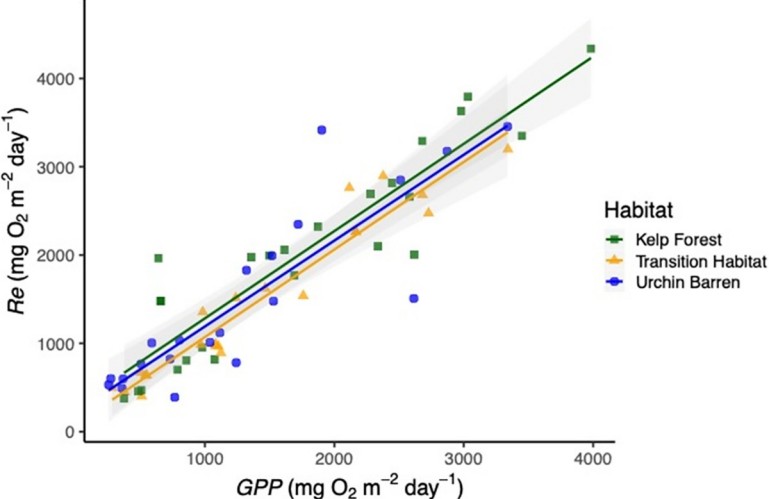

**Fig 6. *GPP* versus *Re*.** Relationship between gross primary production (*GPP*) and ecosystem respiration (*Re*) for each habitat type across all nine islands where cBITs were deployed in 2016 and 2017 (Table 1). Each point represents measurements from a single cBIT. Gray shading denoted 95% confidence intervals.

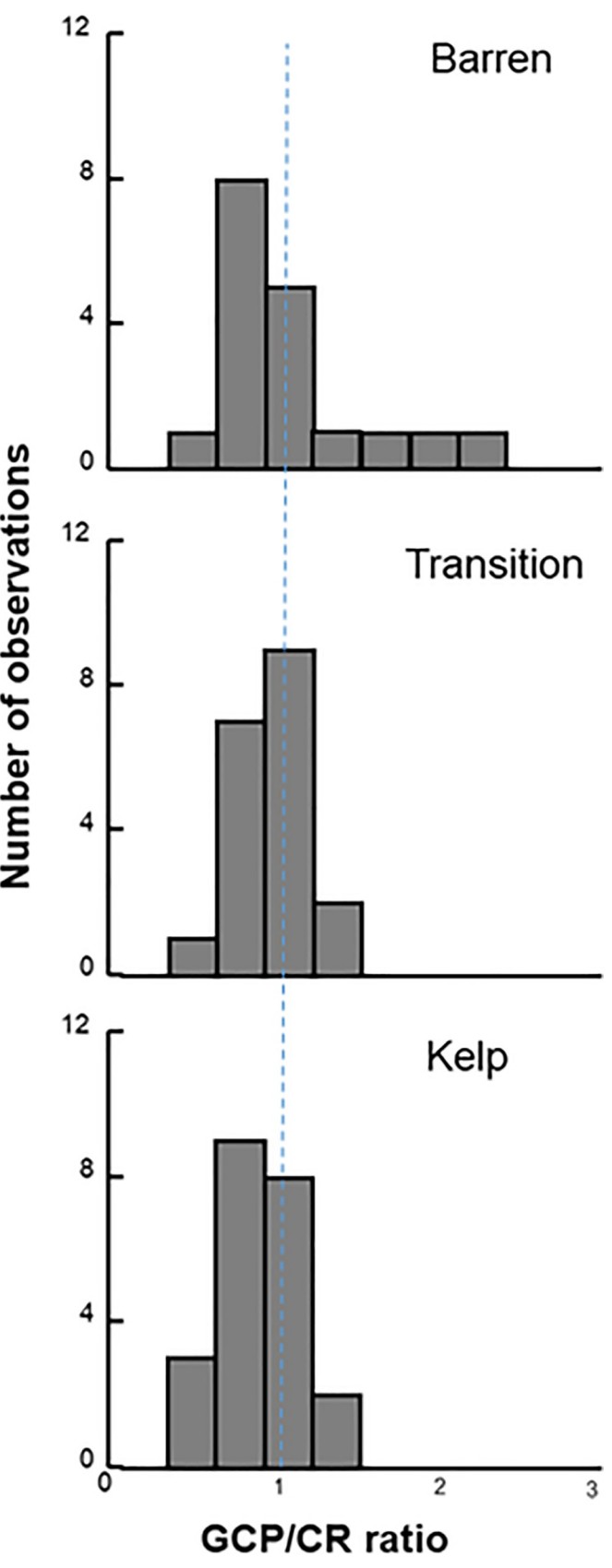

**Fig 7. GPP and Re ratios.** Frequency distribution of *GPP / Re* ratios within each habitat type across all nine islands where cBITs were deployed in 2016 and 2017 (Table 1). Each data point represents measurements from a single cBIT. Note the urchin barrens have the highest ratios observed, and the kelp forests have the largest number of low values. The vertical dashed line represents the 1:1 ratio.

the urchin barrens. The range was between *GPP* and *Re* was 24% greater, on average, in the kelp forests than in the urchin barrens, and 19% greater in the kelp forests than in the transition habitats, but it varied by less than 6% between the transition habitats and the urchin barrens. In contrast, *GPP* was, on average, greatest in the urchin barrens, intermediate in the transition habitats, and lowest in the kelp forests in contrast, but it did not differ from zero (i.e. *GPP* = *Re*) in any of the habitats. These patterns, however, were highly variable among the different islands visited in the two study years for each of the production metrics. Altogether, this indicated deforestation resulted in widespread but geographically variable losses to primary production and respiration by the ecosystem.

As with previous studies in aquatic ecosystems, we found that *GPP* and *Re* are generally in balance, resulting in exhibit *GPP / Re* ratios near 1.0, and *NEP* values near zero [21,22]. When examined within each cBIT separately, *GPP* and *Re* were consistently similar in magnitude with no differences in *GPP / Re* ratios among habitat types (ANCOVA: $F_{2,62}$ = 0.16, p = 0.852) (Table 5, Fig 6). Further, the distribution of these ratios was symmetrical around 1.0 in each habitat (Fig 7). Interestingly, the highest individual values of *NEP* were not observed in the kelp forests but rather in the urchin barrens, which we believe was due to higher irradiances in the urchin barrens than the other two habitats (Fig 5) combined with compensatory production by the encrusting coralline algae and benthic diatoms [30]. However, those few observations aside, it is clear that all three benthic habitats remain in balance following deforestation, with *GPP* ≈ *Re*, *GPP / Re* ratios ≈ 1, and median *NEP* values ≈ 0. Thus, although *NEP* may help differentiate between productive and unproductive ecosystems [22], it poorly describes changes in primary productivity following large-scale habitat change in the Aleutian Archipelago. Instead, it is clear that deforestation results in significant changes to the region's benthic communities, and these led to geographically variable reductions in *GPP*, *Re* and the range between them, which better reflect a reduction in ecosystem functioning. Further, it appears that even partial deforestation, where the benthic macroalgae and invertebrates have been lost but the canopy-forming kelps remain, results in lower *GPP* and *Re* at the benthos that is similar to trends found in urchin barrens.

## Discussion

Trophic interactions can lead to changes in the distribution and abundance of habitat-forming species, which can have profound impacts on ecosystem function [2,9]. Deforestation, in particular, can result in changes to biodiversity and energy flow [2], altered regional and global climates [34], and even lead to species extinctions [35]. Coastal kelps are a pertinent example of such ecosystem engineers in nearshore habitats that have suffered declines in some locations over the past few decades due to both biological and physical stressors [10,11,36–39]. Consequently, while kelp populations remain stable in many of the world's ecoregions [10,40,41], or may even be expanding in some high latitude regions in response to ocean warming [39,42], our study is relevant to many areas of the world where kelp forests have exhibited local to broad scale declines [10,43–47]. Indeed, recent estimates suggest that global declines in kelp abundances may be as high as 2% per year [11], which can negatively affect numerous other species that depend on them for food and habitat. Certainly, the kelp forests of the Aleutian Archipelago are in critical condition in the face of widespread overgrazing by urchins, and this

**Table 3. Community productivity values (measured as mg O$_2$ m$^{-2}$ day$^{-1}$) for A) Net ecosystem production (*NEP*), B) gross primary production (*GPP*), C) ecosystem respiration (*Re*), and D) the Range between *GPP* and *Re* (*Range*) estimated for each habitat on each island.** Data reflect the means (SD) of the replicate chambers in each habitat (kelp Forests, Urchin Barrens and Transition Habitats). Positive values for *NEP* reflect net oxygen production and negative values reflect oxygen consumption (net respiration). Negative values for *Re* reflect oxygen consumption (i.e. more negative values reflect greater respiration by the ecosystem). "NA" denotes not available due to lack of replication (i.e. data are based on only one chamber at that island; see Table 1).

| *NEP* | | | |
|---|---|---|---|
| **Island** | **Kelp** | **Barren** | **Transition** |
| Adak | -851.56 (409.34) | -431.10 (184.56) | -580.85 (89.91) |
| Amchitka | -694.81 (217.86) | -166.91 (141.38) | -151.16 (86.17) |
| Atka | -434.43 (72.69) | -231.05 (137.95) | -208.92 (143.52) |
| Attu | -176.96 (167.98) | 1104.59 (NA) | 98.52 (151.30) |
| Chuginadak | 37.74 (15.07) | -537.29 (80.57) | 105.85 (9.02) |
| Kiska | 193.70 (93.03) | 418.16 (58.16) | 160.13 (63.01) |
| Nizki | -608.20 (177.58) | -20.29 (102.59) | -84.82 (78.95) |
| Tanaga | 355.08 (361.67) | -909.49 (854.06) | 254 (NA) |
| Yunkasa | 21.82 (21.50) | -307.36 (96.49) | -129.13 (132.22) |
| Average | -239.74 | -120.08 | -59.6 |
| SE | 245.47 | 338.07 | 145.32 |
| *GPP* | | | |
| **Island** | **Kelp** | **Barren** | **Transition** |
| Adak | 2018.08 (934.63) | 2296.04 (1164.46) | 2246.55 (184.47) |
| Amchitka | 977.21 (552.29) | 807.74 (304.87) | 416.49 (188.14) |
| Atka | 2450.84 (1340.11) | 315.90 (60.74) | 675.59 (266.57) |
| Attu | 2238.97 (479.65) | 2445.60(NA) | 1826.29 (1482.52) |
| Chuginadak | 917.57 (86.83) | 1519.57 (199.70) | 1081.44 (3.13) |
| Kiska | 1399.85 (823.96) | 1003.82 (335.91) | 912.65 (347.63) |
| Nizki | 2763.69 (420.47) | 1130.44 (564.73) | 2104.32 (610.25) |
| Tanaga | 3032.82 (588.61) | 2386.64 (684.53) | 2727.91 (NA) |
| Yunkasa | 456.27 (71.52) | 404.17 (169.92) | 1456.75 (627.00) |
| Average | 1806.14 | 1367.77 | 1494.22 |
| SE | 521.75 | 483.99 | 452.41 |
| *Re* | | | |
| **Island** | **Kelp** | **Barren** | **Transition** |
| Adak | -2410.85 (763.16) | -2439.14(1273.10) | -2827.40 (94.57) |
| Amchitka | -1672.02 (334.43) | -974.65 (184.45) | -567.65 (101.96) |
| Atka | -2885.26 (1268.85) | -546.95 (77.21) | -884.52 (406.-6) |
| Attu | -2415.93 (564.65) | -1509.01 (NA) | -1727.77 (1313.82) |
| Chuginadak | -879.83 (101.90) | -2056.86 (265.44) | -975.60 (12.15) |
| Kiska | -1206.15 (775.20) | -585.66 (277.75) | -753.52 (308.09) |
| Nizki | -3371.99 (594.26) | -1150.73 (462.14) | -2189.14 (533.21) |
| Tanaga | -2677.74 (950.28) | -3296.13 (169.53) | -2473.12 (NA) |
| Yunkasa | -434.45 (50.10) | -711.43 (256.39) | -1585.88 (641.96) |
| Average | 1994.91 | 1474.51 | 1553.84 |
| SE | 574.11 | 546.83 | 469.81 |
| *Range* | | | |
| **Island** | **Kelp** | **Barren** | **Transition** |
| Adak | 3969.73 (1770.87) | 4735.18 (2432.99) | 5073.94 (279.04) |
| Amchitka | 2649.23 (886.72) | 1782.39 (483.68) | 984.15 (290.10) |
| Atka | 5336.10 (2608.91) | 862.85 (16.48) | 1560.11 (671.78) |
| Attu | 4654.89 (1033.53) | 4122.62 (NA) | 3554.07 (2864.05) |

*(Continued)*

**Table 3.** (Continued)

| Chuginadak | 1797.40 (188.72) | 3576.43 (462.80) | 2057.03 (15.29) |
| Kiska | 2606.00 (1597.22) | 1589.48 (613.66) | 1665.17 (653.88) |
| Nizki | 6135.68 (1014.08) | 2281.17 (1026.88) | 4293.65 (1143.33) |
| Tanaga | 5710.56 (1538.90) | 5682.77 (515.01) | 5201.03 (NA) |
| Yunkasa | 890.72 (121.60) | 1115.60 (424.16) | 3042.63 (1262.13) |
| Average | 3750.03 | 2860.94 | 3047.98 |
| SE | 1069.01 | 994.44 | 910.94 |

has had profound effects on the region's benthic communities and on patterns of gross primary production and ecosystem respiration. Whether these forests will recover and return to prior ecosystem functioning regarding these metrics is unknown, but observations of kelp forests from other areas of the world suggest it is possible. For example, *Laminaria longicruris* forests recovered from overgrazing following localized disease outbreaks that decimated sea urchin populations in Nova Scotia [48], while *L. hyperborea* forests recovered in mid-Norway due to low sea urchin recruitment [49]. *Ecklonia maxima* expanded its range eastward in South Africa, coincident with cooling of the local ocean waters [50]. Likewise, *Macrocystis pyrifera* recovered along a ~100 km stretch of the Pacific coast of Baja California, Mexico following nearly two decades of absence after the strong 1997–98 El Niño Southern Oscillation [51]. Recovery of the *Eualaria fistulosa* forests throughout the Aleutian Archipelago, however, would likely require widespread mortality in the urchin populations, which today seems unlikely. One potential contributing factor for this may lie in the low abundance of other urchin predators, such as the urchin eating sea star, *Pycnopodia helianthoides* [38,52,53], which historically has not been found in high abundances in the central or western Aleutians. Therefore, until predation on the urchins recovers or the urchin populations suffer widespread disease that reduces their numbers, benthic algal abundances, *GPP* and *Re* will likely remain generally lower in areas of kelp forest loss because the high abundance of urchins limits regrowth of macroalgae and maintains the urchin barrens [15]. Thus, we present a benchmark against which we can evaluate this recovery if it occurs, and understand the effects of further deforestation in this ecosystem.

Although we have learned much about the effects of the otter-urchin-kelp trophic cascade in the Aleutian Archipelago, this study offers new insights into the consequences of such widespread deforestation on the region's benthic primary productivity. Certainly, benthic *GPP*, *Re* and the range between them are generally greatest in the kelp forests where macroalgae, fish, invertebrates, and microbial communities are all most abundant [15,23–26,33], while they are lowest in the urchin barrens. Deforestation thus resulted in overall reductions in each of these metrics, identifying a general loss of ecosystem function. This, however, was geographically variable, with some islands showing elevated primary production following deforestation, which we believe is due to higher irradiances combined with compensatory production by microalgae (e.g. diatoms) and the coralline algal crusts. Indeed, we observed some of the highest production values in a few of the barrens cBITs where diatom mats formed within the chambers during the deployments. These cBITs also tended to have low numbers of urchins within them, and the chambers therefore appeared to exclude urchins from grazing the microalgae. In contrast, benthic primary productivity and respiration by the ecosystem are all similar in the urchin barrens and transition habitats, which have similarly high abundances of urchins and low biomasses of macroalgae [15,33], suggesting that the transition habitats have already suffered reduced ecosystem functioning. This, of course, reflects patterns at the benthos and

**Table 4. The effects of habitat change on patterns of productivity for A) Net ecosystem production (*NEP*), B) gross primary production (*GPP*), C) ecosystem respiration (*Re*), and D the Range between *GPP* and *Re* (i.e. Range) estimated for reach island.** "Change" reflects absolute differences in each metric (measured as mg $O_2$ m$^{-2}$ day$^{-1}$) as the habitat transitions from Kelp forests to Transition Habitats, Transition Habitats to Urchin Barrens, Kelp Forests to Urchin Barrens (i.e. the total change due to deforestation). Positive values denote greater values for that metric and negative values denote lower values for that metric. NA denotes comparison "not available" due to loss of replicates in one habitat that precluded reliable estimates of the change (see Table 1). At the bottom of each table are the average values and standard errors.

| A) *NEP* | Kelp to Transition | Transition to Barren | Kelp to Barren |
|---|---|---|---|
| Island | Change | Change | Change |
| Adak | 270.71 | 808.46 | 420.46 |
| Amchitka | 845.97 | -318.07 | 527.9 |
| Atka | 225.51 | -22.13 | 203.38 |
| Attu | 275.48 | NA | NA |
| Chuginadak | 68.11 | -643.14 | -575.03 |
| Kiska | -33.57 | 258.03 | 224.46 |
| Nizki | 523.38 | 64.53 | 587.91 |
| Tanaga | NA | NA | -1264.57 |
| Yunkasa | -150.95 | -178.23 | -329.18 |
| Average | 253.08 | -4.36 | -25.58 |
| SE | 183.33 | 265.76 | 373.26 |

| B) *GPP* | Kelp to Transition | Transition to Barren | Kelp to Barren |
|---|---|---|---|
| Island | Change | Change | Change |
| Adak | 687.67 | 49.49 | 737.16 |
| Amchitka | -560.72 | 391.25 | -169.47 |
| Atka | -1775.25 | -359.69 | -2134.94 |
| Attu | -412.68 | NA | NA |
| Chuginadak | 163.87 | 438.13 | 602 |
| Kiska | -487.2 | 91.17 | -396.03 |
| Nizki | -659.37 | -973.88 | -1633.25 |
| Tanaga | NA | NA | -646.18 |
| Yunkasa | 1000.48 | -1052.58 | -52.1 |
| Average | -255.4 | -202.3 | -461.6 |
| SE | 501.77 | 353.95 | 578.69 |

| C) *Re* | Kelp to Transition | Transition to Barren | Kelp to Barren |
|---|---|---|---|
| Island | Change | Change | Change |
| Adak | 416.55 | -388.26 | 28.29 |
| Amchitka | -1104.37 | 407 | -697.37 |
| Atka | -2000.74 | -337.57 | -2338.31 |
| Attu | -688.16 | NA | NA |
| Chuginadak | 95.77 | 1081.26 | 1177.03 |
| Kiska | -452.63 | -167.86 | -620.49 |
| Nizki | -1182.85 | -1038.41 | -2221.26 |
| Tanaga | NA | NA | 618.39 |
| Yunkasa | 1151.43 | -874.45 | 276.98 |
| Average | -470.63 | -188.33 | -472.09 |
| SE | 578.86 | 423.11 | 734.7 |

| D) *Range* | Kelp to Transition | Transition to Barren | Kelp to Barren |
|---|---|---|---|
| Island | Change | Change | Change |
| Adak | 1104.21 | -338.76 | 765.45 |
| Amchitka | -1665.08 | 798.24 | -866.84 |
| Atka | -3775.99 | -697.26 | -4473.25 |
| Attu | -1100.82 | NA | NA |

*(Continued)*

**Table 4.** (Continued)

| | | | |
|---|---|---|---|
| Chuginadak | 259.63 | 1519.4 | 1779.03 |
| Kiska | -940.83 | -75.69 | -1016.52 |
| Nizki | -1842.03 | -2012.48 | -3854.51 |
| Tanaga | NA | NA | -27.79 |
| Yunkasa | 2151.91 | -1927.03 | 224.88 |
| Average | -726.13 | -390.51 | -933.69 |
| SE | 1073.93 | 754.56 | 1262.65 |

not in the mid-water or at the surface where the canopy-forming *Eualaria fistulosa* remains abundant in the transition habitats. It is likely that these canopy-forming macroalgae would enhance *GPP* and perhaps result in positive values of *NEP* in the mid-water and at the surface in both the kelp forests and transition habitats. However, at the benthos, *GPP* and *Re* remain in balance following deforestation, leading to similar, near-zero *NEP* in all three habitats. We believe this reflects balance between the autotrophic and heterotrophic components of the ecosystem. Specifically, the macroalgae exhibit positive *GPP* as they photosynthesize, grow and increase in abundance, but this results in a concomitant increase in heterotrophic metabolism, which enhances *Re*. In the face of deforestation, both *GPP* and *Re* are reduced, resulting in little to no changes in *NEP*. Thus, we propose that, *GPP*, *Re* and the range between them are better measures of changes to primary productivity than *NEP*. Combining these with estimates of macroalgal and invertebrate diversity and abundance revealed that the Aleutian Archipelago suffered geographically variable losses to ecosystem function following widespread deforestation.

## Materials and methods

While many past experiments examining primary production by autotrophic communities have relied on laboratory experiments that do not incorporate natural fluctuations in abiotic conditions, recent studies have identified techniques that measure primary production *in situ*, thereby increasing the ecological realism of their experiments [54–57]. For example, *in situ* chamber designs have been developed for estimating primary production by individual species [55,56] and whole benthic communities [29,56,57]. In general, estimates of net ecosystem production (*NEP*) on the benthos can be made by measuring changes in dissolved oxygen within chambers that are placed *in situ* over macroalgae and invertebrate communities. In this study, we deployed collapsible benthic isolation tents (cBITs) modelled after those described by Haas et al. [58] and Calhoun et al. [59] that directly measured *in situ* benthic oxygen production and allowed us to estimate gross primary production (*GPP*), ecosystem respiration (*Re*) and net ecosystem production *NEP* by the benthic communities [28,29,55]. These cBITs were the same ones used by Sullaway and Edwards [60] to measure loss of primary productivity following the

**Table 5. Analysis of covariance testing the effect of GPP and habitat on Re.** Note the non-significant Habitat*GPP interaction hat shows no differences in the slopes (i.e. relationships) between GPP and Re among the three habitat types. See Fig 5 for graphical representation.

| Source | Type III SS | df | MS | F-ratio | p-value |
|---|---|---|---|---|---|
| *GPP* | 8.460E+03 | 1 | 8.50E+03 | 3.20E+02 | 0.001 |
| HABITAT | 20.791443 | 2 | 1.00E+01 | 3.90E-01 | 0.68 |
| HABITAT*GPP | 8.6140845 | 2 | 4.30E+00 | 1.60E-01 | 0.852 |
| Error | 1.66E+03 | 62 | 2.70E+01 | | |

displacement of native giant kelp, *Macrocystis pyrifera*, by the invasive *Sargassum horneri* on Catalina Island, CA. Further, because our cBITs encompassed whole benthic communities, species interactions (e.g., shading), and invertebrate and microbial respiration were incorporated into production measurements. These interactions are often not captured in laboratory experiments but are pertinent to understanding *GPP*, *Re*, and *NEP* [61].

## Experimental design

Our cBITs were made from 0.106 cm polycarbonate plastic triangle sheets glued to fiberglass-reinforced vinyl panels (Fig 8). The frames were reinforced using stainless steel tubes with stainless steel cable to facilitate handling and to ensure they held their pyramidal shape with an internal volume of 192 L and a basal area of (0.64 m$^2$). The cBITs each had 26" skirts around the perimeter, upon which chain was laid to hold them to the benthos and prevent water exchange with the surrounding environment. This was verified by injecting fluorescein dye into the chambers and examining the perimeters for leaks. The polycarbonate walls were thin and flexible to allow hydrodynamic energy transfer into the cBITs, thereby reducing boundary layer formation around the macroalgal thalli. We verified this energy transfer using dissolving plaster blocks placed within cBITs, and by using video analysis of internal seaweed and fluorescein dye movements within the chambers relative to seaweeds outside them [60]. Sensor arrays that included a Photosynthetic Active Radiation (*PAR*) sensor (Odyssey Dataflow Systems Ltd), and a Dissolved Oxygen (DO mg/L) and Temperature (°C) sensor (MiniDOT Logger, PME) were placed at the center of each cBIT (Fig 8).

During two cruises aboard the *R/V Oceanus* in 2016 and 2017, we deployed cBITs in each of the three habitats (kelp forest, urchin barrens, transition habitats) on each of nine islands (Table 1; Figs 1,2 and 8) for 36-hour periods to measure both day and night patterns of *NEP* and *Re*, and to ensure we captured a complete diurnal cycle. These islands span more than 1000 km and therefore experience differences in temperature, salinity, wave exposure and other biotic factors [62]. Consequently, all cBITs deployments were done in the summer (i.e. July) of each year, in similar depths (i.e. 6–8 m), and under similar wave exposures (i.e. protected from ocean swells) in order to standardize factors that could affect productivity measurements. The three habitat types were selected based on non-overlapping community assemblages (i.e., kelp forests were chosen based on abundant *E. fistulosa* and dense assemblages of understory macroalgae; transition habitats were chosen based on abundant *E. fistulosa*, little-to-no understory macroalgae, and high abundances of urchins; urchin barrens were chosen based on no *E. fistulosa*, little-to-no understory macroalgae and abundant urchins). These were then grouped in each island to reduce the effects within-island spatial heterogeneity in other environmental factors. For each deployment, three replicate cBITs were placed on the benthos over targeted assemblages within each habitat type. However, occasionally, replicates were lost due to logistical difficulties associated with the chamber-benthos seals (Table 1). The water within each cBIT was replaced once per day by opening the side of the chamber and completely replacing the water with new ambient seawater to reduce "chamber effects" (i.e., the build-up of oxygen and depletion of inorganic carbon and nutrients). After each deployment, the chambers and sensors were retrieved. At six of the islands (Table 1), all organisms within each of the chambers' benthic footprints were collected, brought back to the ship, enumerated and weighed during our 2017 cruise. We measured *NEP* over the whole diurnal cycle, *Re* during the nighttime hours, and calculated *GPP* during the day for each cBIT during each incubation period separately according to Olivé et al. [57]. Specifically, measurements made during the night (the dark) were used to infer rates of *Re*, which were then combined with measurements of *NEP* to estimate *GPP* by the autotrophs [18–20]. Ethical

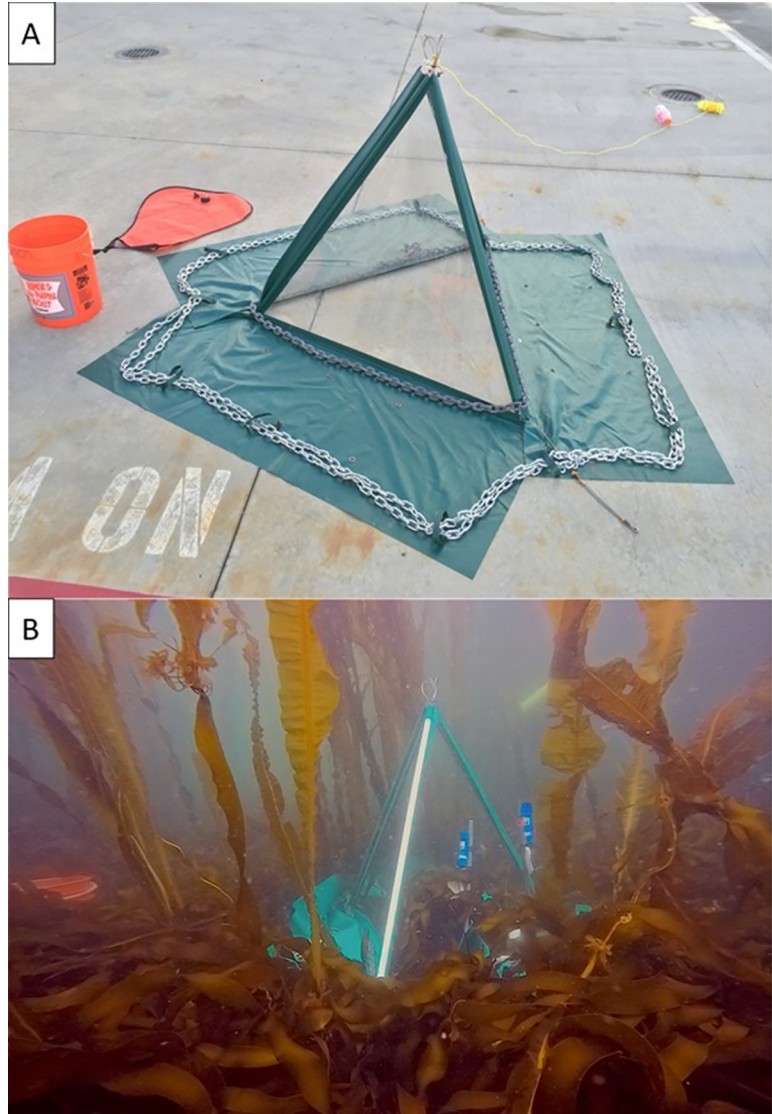

**Fig 8. Photograph of cBIT.** Photograph of (A) cBIT before deployment showing 26" skirt around perimeter, flexible polycarbonate walls, steel framing, anchor chain used to hold skirt and cBIT to the benthos, and (B) cBIT deployed in kelp forest showing *PAR* and oxygen sensors placed both inside and outside the chamber.

Approval: All procedures performed in studies involving fishes were in accordance with the ethical standards of the institution or practice at which the studies were conducted (University of Alaska Fairbanks Institutional Animal Care and Use Committee; Permit Number: 899401–4).

## Statistical analyses

All analyses were done in either Systat ver. 12 or Primer ver 6. Prior to analyses, all data were evaluated for normality by graphical examination of the residuals, which suggested they were slightly non-normal. Data for *NPP*, *GPP*, *Re* and the Range between *GPP* and *Re* were then square-root transformed and re-graphed, which suggested the problems were corrected. Data for *PAR* were log transformed, which corrected the problem. The transformed data were then

examined for equality of variances using Bartlette's tests, which indicated they were homosce-dastic. We then evaluated if urchin biomass, *PAR*, *GPP*, *Re*, *NEP* and the range between *GPP* and *Re* varied among the three habitats (kelp forests, urchin barrens, and transition habitats), the nine islands, and between the two study years using separate three-way Model III Nested ANOVAs, with year and habitat type as fixed factors, island nested within year as a random factor. We evaluated if the relationship between *GPP* and *Re* varied among habitats using ANCOVA, with *Re* as the response variable, *GPP* as the covariate, and habitat type as the cate-gorical independent variable. We evaluated if the ratios in any of the habitats differed from 1.0 (i.e. *GPP* = *Re*) by assessing if the value 1.0 occurred within the 95% confidence intervals around their average values.

## Acknowledgments

We thank S. Lamerdin, and the captain and crew of the *R/V Oceanus* for excellent ship sup-port. We thank J. Estes for offering historical perspectives on the Aleutian kelp ecosystem, and M. Hatay for designing the cBITs. We are grateful to M. Good, S. Traiger, J. Metzger, A. Bland, A. Ravelo, and B. Weitzman for assistance with field operations. We also thank the Alaska Maritime National Wildlife Refuge for logistical support.

## Author Contributions

**Conceptualization:** Matthew Edwards, Brenda Konar.

**Data curation:** Matthew Edwards, Michael Spector.

**Formal analysis:** Matthew Edwards, Scott Gabara.

**Funding acquisition:** Matthew Edwards, Brenda Konar, Ju-Hyoung Kim.

**Investigation:** Matthew Edwards, Brenda Konar, Ju-Hyoung Kim, Scott Gabara, Genoa Sull-away, Tristin McHugh, Michael Spector, Sadie Small.

**Methodology:** Matthew Edwards, Brenda Konar, Ju-Hyoung Kim.

**Project administration:** Matthew Edwards, Brenda Konar.

**Supervision:** Matthew Edwards.

**Validation:** Ju-Hyoung Kim.

**Visualization:** Scott Gabara.

**Writing – original draft:** Matthew Edwards.

**Writing – review & editing:** Matthew Edwards, Brenda Konar, Ju-Hyoung Kim, Scott Gabara, Genoa Sullaway, Tristin McHugh, Michael Spector, Sadie Small.

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
