## [Decision Letter · Decision Letter 0]

2 Jan 2020

PONE-D-19-32311

Marine deforestation leads to widespread loss of ecosystem function

PLOS ONE

Dear Dr. Edwards,

Thank you for submitting your manuscript to PLOS ONE. After careful consideration, we feel that it has merit but does not fully meet PLOS ONE’s publication criteria as it currently stands. Therefore, we invite you to submit a revised version of the manuscript that addresses the points raised during the review process.

We would appreciate receiving your revised manuscript by Feb 16 2020 11:59PM. To enhance the reproducibility of your results, we recommend that if applicable you deposit your laboratory protocols in protocols.io, where a protocol can be assigned its own identifier (DOI) such that it can be cited independently in the future. For instructions see: http://journals.plos.org/plosone/s/submission-guidelines#loc-laboratory-protocols

We look forward to receiving your revised manuscript.

Kind regards,

Maura (Gee) Geraldine Chapman, PhD DSc

Academic Editor

PLOS ONE

Journal Requirements:

3. Our internal editors have looked over your manuscript and determined that it is within the scope of our Biodiversity Conservation Call for Papers. This collection of papers is headed by a team of Guest Editors for PLOS ONE (https://collections.plos.org/s/biodiversity). The Collection will encompass a diverse range of research articles on biodiversity conservation, including the impacts of deforestation and habitat loss. Additional information can be found on our announcement page: https://collections.plos.org/s/biodiversity

If you would like your manuscript to be considered for this collection, please let us know in your cover letter and we will ensure that your paper is treated as if you were responding to this call. If you would prefer to remove your manuscript from collection consideration, please specify this in the cover letter.

"This research was funded by grants from the National Science Foundation (OCE1435194) to MSE and BK, and the National Research Foundation (NRF-2018R1C1B6008523 and NRF-2015R1C1A1A01054831) to JHK."

"NO - The funders had no role in study design, data collection and analysis, decision to publish, or preparation of the manuscript."

5. We note that Figure #2 in your submission contains map images which may be copyrighted. All PLOS content is published under the Creative Commons Attribution License (CC BY 4.0), which means that the manuscript, images, and Supporting Information files will be freely available online, and any third party is permitted to access, download, copy, distribute, and use these materials in any way, even commercially, with proper attribution. For these reasons, we cannot publish previously copyrighted maps or satellite images created using proprietary data, such as Google software (Google Maps, Street View, and Earth). For more information, see our copyright guidelines: http://journals.plos.org/plosone/s/licenses-and-copyright.

a.    You may seek permission from the original copyright holder of Figure(s) [#] to publish the content specifically under the CC BY 4.0 license.

6. Please upload a copy of Supporting Information Table S1 which you refer to in your text on page 23.

Reviewers' comments:

Reviewer's Responses to Questions

**Comments to the Author**

1. Is the manuscript technically sound, and do the data support the conclusions?

Reviewer #1: Partly

Reviewer #2: Yes

Reviewer #3: Yes

2. Has the statistical analysis been performed appropriately and rigorously? 

Reviewer #1: Yes

Reviewer #2: Yes

Reviewer #3: Yes

3. Have the authors made all data underlying the findings in their manuscript fully available?

Reviewer #1: Yes

Reviewer #2: Yes

Reviewer #3: Yes

4. Is the manuscript presented in an intelligible fashion and written in standard English?

Reviewer #1: Yes

Reviewer #2: Yes

Reviewer #3: Yes

5. Review Comments to the Author

Reviewer #1: The authors report on productivity estimates in the nearshore of the Aleutians, studying a community that has likely changed as a result of orca whale predation on sea otters, thus leading to high urchin densities in some areas. The authors use chambers in situ to measure productivity across habitat types that differ: kelp-dominated versus crustose coralline-dominated. The measurements they have made in the field with these chambers are valuable and include the whole in situ community. The effort to do this was significant. I thus think this is a very worthwhile study, though I have several comments to increase the clarity of the presentation and interpretation.

First, I was surprised not to see any mention of seastar die-off as a driver of high urchin abundances. Are the findings of Burt et al. 2018 (PNAS) in Canada irrelevant here? Are there any data for seastar die-offs?

Second, and importantly, parts of the Discussion contradict the Results. For example, at L 230 it is stated that GPP and Re will remain lower in areas of kelp forest loss. Yet, Figs show Re is higher. Again on L 239, L252 In fact, the Figs show increased GPP and decreased Re in kelp forests. Is there a mistake on your figs or in the text explanation? Or did I miss something? Even though the data were analyzed with a mixed effects model using island as a random effect, the fact that islands differ make this difficult to interpret. In general, the explanations that the reader gets in the Results and Discussion do not match the complexity of the data.

L46 unnecessary text: “Nowhere may this be as dramatic as..”

L112 canopy algae also host a diversity of microbes directly (Michelou 2013 PlosOne, Weigel and Pfister 2019 Front in Microb).

L121 “Understanding the balance…to NEP is a key aspect to understanding how habitat-driven ecosystem change occurs. We test the relationship among GPP, Re, NEP in the Aleutian Archipelago where recent widespread …” I though this section needed a much more direct statement.

L146 Benthic irradiances as photosynt active rad (PAR)…

L196 and in discussion: perhaps replace ‘widespread kelp deforestation’ with ‘large-scale habitat change’. The primary producers have changed and that is your point

L263, 264 – I recall coral reef in situ chambers by Williams and Carpenter 1998 that came much before the cited papers.

Table 2 – I’m glad the authors used island as a random factor

Table 4 – what do interactions tell us? Were they discussed? Are these interactions important in the confusing reporting of GPP and Re trends?

Reviewer #2: Edwards et al. provide a novel approach to quantifying the effects of trophic downgrading on kelp forest ecosystems. By empirically quantifying ecosystem rates across high latitude temperate reefs varying in macroalgal biomass, this manuscript offers a unique and important contribution to our understating of kelp forest dynamics globally. The strength of this paper lies in the collection of these empirical rates. Moreover, the monumental field effort to acquire these empirical rate data across 1000km is remarkable and these hard earned and unique data should be reported in the literature.

There are also several key weaknesses to this paper that the authors could address by fine tuning the writing, hypotheses and quantitative analyses. First, the analyses and figures do not tell the story the manuscript’s introduction and hypotheses aim to set up. Specifically, the alternative hypotheses presented relate GPP, Re and NEP to variation in primary producer biomass driven by the deforestation of kelp by sea urchins (Lines 114- 123, 155-156). Moreover, the authors suggest that ‘complex patterns’ of these rates may be driven by variation in macroalgal biomass, irradiance and secondary producer biomass and biodiversity. However, the quantitative models and the graphs themselves never assess these relationships and plausible causal mechanisms. Instead, habitat was used as a categorial variable and proxy for primary producer biomass. An alternative approach would be to run a set of linear models with primary producer biomass as a covariate, along with other important covariates implicated by the authors but not tested (Lines 116-121). The authors could also choose to keep their analyses as is yet better articulate and illustrate their hypotheses under various categorical scenarios of biomass, irradiance and secondary production. When alternative hypotheses and the variables they invoke match the empirical data collected, tested and presented, it makes it much easier for future readers to understand the evidence in support of alternative hypotheses.

Second, as the authors know well, factors beyond herbivory are well known to influence kelp forest structure and function. The manuscript would be stronger if these additional drivers of change were acknowledged in the text and addressed in the quantitative analysis. Specifically, the analyses considered ‘island’ as a random factor and ‘habitat’ as a fixed effect. Yet, across 1000km, surely these islands, and the habitats nested within them, varied in some biotic and abiotic conditions other than herbivory. This makes it difficult to treat islands as quantitative ‘replicates’. Did islands and habitats where the rates were collected also vary in wave exposure, depth, nutrients and sea water temperature? Might these factors have influenced the ecosystem rates measured? I would think so to some degree over this massive geographical area. Perhaps the authors chose sites to reduce these variables. If so, this should be clearly stated. At most this could be tested, at least this should be acknowledged.

Third, the quantitative approach seems a bit awkward given the hypotheses tested. Why were permutation tests done on transformed data when a simple generalized linear model with an appropriate link function be more parsimonious? There may be a good reason for this, it just wasn’t clear to me. It would be helpful if the authors justified their quantitative approach in the text. Were the 3 replicates done in the same location within a habitat type nested within an Island?

Minor Points

Data collection was done in 2016 and 2017, but the authors do not specify which sites were sampled in what year. As temperature and other anomalies vary between years, did the authors test for an effect of year in their analysis?

The authors’ use of the term ‘ecosystem health’ is confusing because it obfuscates what is inferred from what was measured. To clarify their inferences, I recommend that the authors consider removing this nondescript term and instead use the word to describe the metric measured.

Biodiversity is mentioned throughout the text but never quantified.

Nowhere in the manuscript is variation in sea otter abundance or occupation time reported and yet the inferences made about deforestation are all related to sea otter depletion.

A substantial amount of inference appears in the results section (ex: lines 161-163, 166-169, 183-201). I would recommend keeping the inference to the discussion section. Moreover, some of the inferences (ex: lines 189-191) made could be tested with data (in this case PAR measurements).

There are inconsistencies in the definitions of NEP (lines 99-100 & 169-171) that make for a circular argument presented in the discussion (lines 174-177).

To improve Fig 4B, consider graphing the encrusting invertebrates in a separate graph with a smaller y axis scale so readers can see the differences in inverts among kelp, transitions sites and urchin barrens. These are great data and I would love to see them better.

Fig 7 and 8: Consider joining them into one figure

Line 105: Two ‘indeeds’

Line 119: Benthic irradiances not defined

Line 130: Define what defines a transition site? Sea otter occupation? Kelp density?

Line 294 – cBITs deployed for 24-36 hour periods. How does variation in deployment time affect results?

Line 169: ‘Lastly, the difference (i.e. range) between GPP and Re, which we believe to be a better measure of ecosystem function than NEP..’ but in lines 99-100 you write: ‘net ecosystem production (NEP),which is the difference between GPP and Re.’

line 179: indicted should be indicated

Lines 238–240: Unclear. Does ecosystem health and function = biodiversity, macroalgal abundances, and primary productivity?

Methods: Please specify depth and season when cBITs were deployed as kelp growth, photoperiod, upwelling intensity, phytoplankton blooms, and wave exposure vary as a function of season, and all of these factors could influence GPP, Re, and NEP.

Line 545 Table 1: It would be nice to see totals for number of CBITs deployed by habitat type. Could also add collection year/month as a column. This Table could be in an Appendix.

Reviewer #3: Review of PONE-D-19-32311 by Edwards et al

The authors present results from an interesting study examining changes in ecosystem functioning following a shift in Alaskan coastal habitats from kelp forests to urchin barrens. The study is well conceived and executed, the paper is well written and the data are analysed and interpreted appropriately. In fact, I really enjoyed reading the paper and think it will be very well received by the marine ecological community. It provides a neat and pertinent example of how structural shifts alter functioning and also raises some interesting points about the usefulness of GPP, NPP and Re measurements in coastal systems. I have some relatively minor concerns and suggestions for improvement that the authors should consider.

Line 66-67: On first reading I found the link between predators and forest-forming trees odd as predators don’t eat trees. I think either change ‘predators’ to ‘consumers’ or insert ‘directly or indirectly’ after ‘they result’

Line 85: The collapse of sea otter populations in the region began a long time before the 1980s due to hunting otters for fur. I think it would be useful to add a little more historical context here for readers unfamiliar with the system. Maybe just clarify that this recent collapse of sea otter numbers comes after a historical collapse and recovery.

Line 88: Somewhere in the introduction, maybe around here, it would be good to provide some details about the kelp forests themselves. I had to read to the discussion before the actual species of kelp was mentioned. Just briefly, what is the general structure of these forests? Does a single species dominate or are a few species important?

Line 91: I think the word ‘excellent’ is slightly subjective. Perhaps ‘useful’ or ‘valuable’ would be more scientific?

Line 208: Similarly, replace ‘excellent’ with useful or pertinent

Line 210-215: I feel this section is a little unbalanced, as it implies that many (most?) kelp forests are in decline, whereas in fact most kelp ecosystems globally are stable and some are even expanding into polar regions. The paradigm that kelp forests are structured by trophic cascades is not true in many (most?) regions and the idea that they are in decline in most regions is also not supported by field observations. I think it should be made explicit that many kelp forests are intact, to provide balance for the statements focussing on kelp loss. For example, Krumhansl et al (2016 PNAS) show that kelps in 62 ecoregions studied were either stable or increasing and Smale (2019 New Phytologist) showed that kelp populations are expanding at high latitudes in response to warming. Finally, some of the references provided in the section don’t really support the statement being made. Specifically Pfister et al (ref 34) showed that kelp populations in the northeast Pacific were by and large stable, and declines were very localised. Similarly, Raybaud et al 2013 (ref 36) examined a single species and showed it has/will decline in France but this doesn’t really support ‘kelp forests’ or ‘Western Europe’. Again, the paper by Martinez et al (ref 39) used species distribution models to project future distributions of seaweeds in southern Australia, they did not show that they have exhibited declines. This may be better rephrased as ‘southwest Australia’ and supported by Wernberg et al 2013 Nat Clim Change or Wernberg et al 2016 Science which did show kelp decline in this region.

Line 306: Which time of year were the incubations conducted? How would seasonal variability in assemblage structure, light or productivity affect the patterns? Are the results representative of the annual cycle? It would be useful to include a statement about the time of sampling and possible caveats of seasonal variability.

Line 321: Typo with ‘PERMNOVA’

6. PLOS authors have the option to publish the peer review history of their article (what does this mean?). If published, this will include your full peer review and any attached files.

Reviewer #1: No

Reviewer #2: No

Reviewer #3: No

---

## [Author Response · Author response to Decision Letter 0]

6 Feb 2020

Please see the response to reviewers. The document is large and better uploaded than inserted here.

---

## [Editor Report · Decision Letter 1]

12 Feb 2020

PONE-D-19-32311R1

Marine deforestation leads to widespread loss of ecosystem function

PLOS ONE

Dear Dr. Edwards,

Thank you for submitting your manuscript to PLOS ONE. After careful consideration, we feel that it has merit but does not fully meet PLOS ONE’s publication criteria as it currently stands. Therefore, we invite you to submit a revised version of the manuscript that addresses the points raised during the review process.

Academic Editor

Thank you for the revision of your manuscript.  Having read your responses and the revised paper, I find no need to send it our for further review.  You have adequately answered all of the reviewers’ comments and produced a good, easy to read paper.  I therefore will accept it for publication, subject to the following very minor revisions.

 L. 96.  “patterns **of** biodiversity”L. 97.  "latter” not “later”L. 381.  Remove “of” after “over”LL. 179-218.  Please change text everywhere where you refer to measurements increasing or decreasing in response to environmental conditions, as my understanding is that you are not referring to temporal measurements that you have made (i.e. before and after deforestation), but spatial measurements (e.g. deforested areas and forested areas, for example).  Therefore, you can only talk about differences, not changes, that require temporal measurements.  Please check throughout the manuscript for this, or similar comments, e.g. “declines” in l. 295, rather than smaller values.

We would appreciate receiving your revised manuscript by Mar 28 2020 11:59PM. To enhance the reproducibility of your results, we recommend that if applicable you deposit your laboratory protocols in protocols.io, where a protocol can be assigned its own identifier (DOI) such that it can be cited independently in the future. For instructions see: http://journals.plos.org/plosone/s/submission-guidelines#loc-laboratory-protocols

We look forward to receiving your revised manuscript.

Kind regards,

Maura (Gee) Geraldine Chapman, PhD DSc

Academic Editor

PLOS ONE

Additional Editor Comments (if provided):

Academic Editor

Thank you for the revision of your manuscript. Having read your responses and the revised paper, I find no need to send it our for further review. You have adequately answered all of the reviewers’ comments and produced a good, easy to read paper. I therefore will accept it for publication, subject to the following very minor revisions.

1. L. 96. “patterns of biodiversity”

2. L. 97. "latter” not “later”

3. L. 381. Remove “of” after “over”

4. LL. 179-218. Please change text everywhere where you refer to measurements increasing or decreasing in response to environmental conditions, as my understanding is that you are not referring to temporal measurements that you have made (i.e. before and after deforestation), but spatial measurements (e.g. deforested areas and forested areas, for example). Therefore, you can only talk about differences, not changes, that require temporal measurements. Please check throughout the manuscript for this, or similar comments, e.g. “declines” in l. 295, rather than smaller values.

---

## [Author Response · Author response to Decision Letter 1]

12 Feb 2020

Below are the comments from the editor and our response below them.

1. L. 96. “patterns of biodiversity”

We have fixed this

2. L. 97. "latter” not “later”

We have done this

3. L. 381. Remove “of” after “over”

We have done this

4. LL. 179-218. Please change text everywhere where you refer to measurements increasing or decreasing in response to environmental conditions, as my understanding is that you are not referring to temporal measurements that you have made (i.e. before and after deforestation), but spatial measurements (e.g. deforested areas and forested areas, for example). Therefore, you can only talk about differences, not changes, that require temporal measurements. Please check throughout the manuscript for this, or similar comments, e.g. “declines” in l. 295, rather than smaller values.

We have replaced all wording that implies rate or temporal changes to verbiage that reflects differences between the habitats. The few places where we left the words decline or increase is where it appropriately reflects temporal changes that have occurred in the ecosystem over the past decades, such as how the forests were lost or the urchin increased over a period of several years. All the verbiage referring to the differences we observed have been amended.

e.g. see lines 42-43, 118-123, 187-192, 200, 207-209, 216-219

---

## [Editor Report · Decision Letter 2]

14 Feb 2020

Marine deforestation leads to widespread loss of ecosystem function

PONE-D-19-32311R2

Dear Dr. Edwards,

We are pleased to inform you that your manuscript has been judged scientifically suitable for publication and will be formally accepted for publication once it complies with all outstanding technical requirements.

With kind regards,

Maura (Gee) Geraldine Chapman, PhD DSc

Academic Editor

PLOS ONE
---

## [Editor Report · Acceptance letter]

19 Feb 2020

PONE-D-19-32311R2 

Marine deforestation leads to widespread loss of ecosystem function 

Dear Dr. Edwards:

I am pleased to inform you that your manuscript has been deemed suitable for publication in PLOS ONE. Congratulations! Your manuscript is now with our production department. 

With kind regards,

on behalf of

Professor Maura (Gee) Geraldine Chapman 

Academic Editor

PLOS ONE